# Does the Low-Carbon City Pilot Policy Promote Green Technology Innovation? Based on Green Patent Data of Chinese A-Share Listed Companies

**DOI:** 10.3390/ijerph18073695

**Published:** 2021-04-01

**Authors:** Jintao Ma, Qiuguang Hu, Weiteng Shen, Xinyi Wei

**Affiliations:** 1School of Business, Ningbo University, Ningbo 315211, China; moshanke@163.com (J.M.); 1801092003@nbu.edu.cn (W.S.); 2001010006@nbu.edu.cn (X.W.); 2East China Sea Institute, Ningbo University, Ningbo 315211, China

**Keywords:** low-carbon city pilot policy, corporate green technological innovation, multi-period DID

## Abstract

To cope with climate change and achieve sustainable development, low-carbon city pilot policies have been implemented. An objective assessment of the performance of these policies facilitates not only the implementation of relevant work in pilot areas, but also the further promotion of these policies. This study uses A-share listed enterprises from 2005 to 2019 and creates a multi-period difference-in-differences model to explore the impact of low-carbon city pilot policies on corporate green technology innovation from multiple dimensions. Results show that (1) low-carbon city pilot policies stimulates the green technological innovation of enterprises as manifested in their application of green invention patents; (2) the introduction of pilot policies is highly conducive to green technological innovation in eastern cities and enterprises in high-carbon emission industries; and (3) tax incentives and government subsidies are important fiscal and taxation tools that play the role of pilot policies in low-carbon cities. By alleviating corporate financing constraints, these policies effectively promote the green technological innovation of enterprises. This study expands the research on the performance of low-carbon city pilot policies and provides data support for a follow-up implementation and promotion of policies from the micro perspective at the enterprise level.

## 1. Introduction

China’s economic development has entered a new normal, but the excessive growth of its carbon emissions has restricted its long-term economic development. To fulfill energy conservation and emission reduction goals while realizing sustainable development, the pilot construction of low-carbon cities came into being. The Chinese government follows the logic of “from point to surface”. Under the authorization of the central government, local governments independently carry out low-carbon city pilot work, and the central government absorbs the best of them into the policies formulated by the central government, and then promotes them nationwide. The first batch of low-carbon pilot projects were launched in 2010. After several years of development, 6 provinces and 81 cities were included in this project. Each pilot attaches great importance to low-carbon development planning and introduced relevant policies based on objective factors, such as resource endowments and industrial structure, with an emphasis on a coordinated economic development and environmental optimization. The construction of low-carbon cities requires an urgent improvement of the existing resource allocation mechanism and a transformation and upgrading of high-energy-consuming industries. As an important entity that gives full play to the role of the market in improving energy efficiency and optimizing energy structure, the innovation of enterprises in the field of green technology plays a key role in transforming economic models. In 2020, low-carbon pilot works were launched nationwide. As a result, the performance of environmental regulations, such as low-carbon city pilot policies, began to attract research attention.

Many researchers at home and abroad have explored the impact of environmental regulation policies on green technology innovation. Most of these studies have analyzed the impact of environmental regulations on green technology innovation based on the “Porter hypothesis.” Given the objective differences in their samples and measurement methods, these studies have proposed three main views. First, environmental regulations inhibit green technological innovation. Some scholars believe that the environmental regulations that increase the cost of pollution control has led to an overall decline in industrial performance [1], squeezed research and development (R&D) investments, and inhibited corporate innovation in the field of green technology [2,3]. For instance, Chintrakarn [4] pointed out that environmental regulations have not promoted the US manufacturing sector. As efficiency is improved, Zhang [5] found that regulated companies mainly purchase new pollution control equipment to reach clean production standards instead of increasing the intensity of their R&D and innovation. Second, environmental regulations promote green technological innovation. Many scholars [6,7,8] find that the “innovation compensation” effect stimulated by environmental regulation policies effectively promotes the innovation of enterprises in the field of green technology. Jing [9] argued that reasonable environmental regulation promotes the low-carbon transformation of industrial enterprises, Jia [10] and Wang [11] pointed out that strengthening environmental regulations can promote green technology innovation, and Qi [12] found that pilot emissions trading policies promote the green innovation activities of enterprises. Third, the impact of environmental regulations on green technology innovation is uncertain. Some studies either show that such relationship is inconsistent [13,14] or point out a non-linear relationship [15,16], and due to differences in their samples and indicators, these studies produce varying conclusions. Yu [17] examined resource-based companies and found an inverted U-shaped relationship between environmental regulations and corporate green technology innovation performance. Using inter-provincial data, Zhang [18] found that the impact of environmental regulations on green technological innovation has a U-shaped characteristic of initially restraining and then promoting. Wang [19] conducted an empirical analysis by using the green patent data of Chinese listed companies and confirmed the aforementioned U-shaped relationship.

In low-carbon cities, low-carbon city pilot policies are used as comprehensive environmental regulation tools, and the effect of their implementation has received much research attention. Many scholars have adopted synthetic control methods or conducted a difference-in-differences (DID) to evaluate the performance of pilot policies and found that the promotion of these policies is conducive to reducing carbon emissions [20,21], decreasing energy consumption [22], and optimizing industrial structure [23,24,25]. Song [26] found that carbon city construction effectively reduces urban air pollution by reducing corporate emissions and upgrading the industrial structure. Zhang [27] found that low-carbon cities reduce carbon emissions by reducing power consumption and improving technological innovation. Many scholars have also examined the performance of low-carbon cities in the development of a green economy. Wang [28] found that low-carbon pilot policies effectively promote the growth of green economies in pilot cities through the causal inference of progressive DID, whereas She [29] further pointed out that low-carbon pilot policies indirectly improve the green total factor productivity of pilot cities by promoting urban innovation and industrial upgrading.

In sum, while previous studies have analyzed the impact of environmental regulations on green technology innovation, they have mainly focused on a single environmental policy as their test object, whereas their evaluation and examination of the pilot policies in low-carbon cities mostly relied on energy consumption as indicators. Although these indicators directly reflect the low-carbon achievements of pilot cities, they are unable to reflect the goal of urban green development. Although some scholars [28,29] have gradually paid attention to this area in recent years, only few have focused on the impact of low-carbon city construction on the green technological innovation of enterprises. Moreover, the micro-mechanism of these policies also needs to be tested. The main innovations of this study are as follows: (1) based on the micro perspective of enterprises, we applied a multi-period DID method to quantitatively evaluate low-carbon city pilot policies, explore their impact on green technology innovation, break through the original macro research paradigm, and expand the policy-related research; (2) on the basis of a benchmark analysis, we further tested the impact of regional heterogeneity and industry carbon intensity on the implementation of low-carbon city pilot policies and explored the direction of these policies; and (3) from the perspective of alleviating financing constraints, we analyzed the impact mechanism of low-carbon city pilot policies on corporate green technological innovation, and offered empirical support for the follow-up implementation and promotion of policies.

## 2. Research Hypotheses

As city-level environmental regulations, low-carbon city pilot policies are not strongly binding. The central government has not set specific targets for local governments in pilot cities, such as for carbon emissions and emission standards. Each pilot city has received many policy incentives and can gradually implement low-carbon work according to its regional development stage and industrial structure characteristics. The introduction of various green financial policies, such as tax incentives and government subsidies, can also guide enterprises to innovate in the field of green technology and control their greenhouse gas emissions. These pilot cities show a certain degree of freedom, but in terms of talent quality, technical foundation, market environment, and geographical location, there are great differences among the pilot areas, local governments also show certain differences in their enthusiasm and innovation in implementing low-carbon policies. These differences in turn influence the progress of related work and the effect of the final implementation. Therefore, in the context of weakly binding policies, whether the establishment of low-carbon cities can promote enterprise innovation in the field of green technology should be tested.

### 2.1. Low-Carbon City Pilot Policies and Corporate Green Technology Innovation

Low-carbon city pilot policies are mainly implemented by reducing energy consumption and emissions in the production process, improving energy efficiency, and promoting the low-carbon transformation of industries. In this process, enterprises as the main innovation players play a vital role. The green production technology developed via scientific research innovation is key to the implementation of relevant policies. On the one hand, environmental regulations will increase the pollution control and emission reduction costs of enterprises and affect their market competitiveness. They may squeeze out these enterprises’ investment in technology R&D, change their financial investment preferences, and drive them away from reality [30]. On the other hand, as pointed out by Porter [31], environmental protection, and economic development are not mutually opposed. Reasonable environmental regulations trigger innovation compensation effects, stimulate corporate green technology innovation, and produce benefits that can make up for the environmental costs of enterprises. Following the above discussion, Hypothesis 1 is proposed as follows.

**Hypothesis** **1** **(H1).**
*Low-carbon city pilot policies are conducive to corporate green technological innovation.*


### 2.2. Low-Carbon City Pilot Policies and Regional Heterogeneity

Prefecture-level cities in different regions often show huge differences in their market environments, resource endowments, traffic conditions, and talent supply, all of which affect their advancement of low-carbon pilot work. Therefore, some regional heterogeneity may be observed in the impact of low-carbon city pilot policies on corporate green technological innovation. Eastern cities in China are generally believed to have more obvious advantages in their accumulation of low-carbon technologies and innovative talents and in their construction of social networks [28,32]. Meanwhile, central and western cities in China not only have weak economic foundation, but also accommodate many energy-consuming and high-pollution enterprises, which lead to severe resource and environmental problems that require an improvement of the overall innovation environment [33]. However, previous studies also show that due to the high population concentration and large economic scale of eastern cities, their carbon lock-in effect is often stronger than that of western cities, and quickly realizing the effectiveness of low-carbon pilot policies is difficult [27]. Although many studies have examined the regional heterogeneity of low-carbon pilot policies [34], most of them have focused on urban technological innovation and carbon emissions [35]. Therefore, the impact of low-carbon pilot policies on the green technology innovation of enterprises across different regions should be examined. From the above discussion, Hypothesis 2 is proposed as follows.

**Hypothesis** **2** **(H2).**
*Low-carbon city pilot policies are more conducive to corporate green technological innovation in eastern cities than in central and western cities.*


### 2.3. Low-Carbon City Pilot Policies and Industry Carbon Intensity

Whether companies are in high- or low-carbon emission industries, they will be affected by environmental regulations, but low-carbon emission industries have strong innovation capabilities, low environmental pollution, and high technology accumulation. Under the regulation of low-carbon city policies, high- and low-carbon companies show significant differences in their green technology innovation. Qi [12] found that compared with clean enterprises, the pilot emission trading policy has a more significant promotion effect on the green technology innovation of polluting enterprises. Shen [36] pointed out that, for different industries, environmental policies have a heterogeneous effect on the green factor productivity of enterprises. According to the policies issued by pilot cities, high-carbon emission industries are often subject to mandatory constraints due to their high-energy consumption and emissions and are strictly restricted in terms of their production capacity, emissions, and energy consumption. These restrictions not only control the entry of new enterprises and the implementation of new projects but also significantly increase the cost of environmental management. To avoid being eliminated, high-carbon emission industries will increase their investments in scientific research and improve their market competitiveness through green technology innovation. In contrast, low-carbon emission industries have received much policy support, and the cost of pollution control has not increased significantly. In addition, they have invariably attached great importance to green technology innovation. Based on the above discussion, Hypothesis 3 is proposed as follows.

**Hypothesis** **3** **(H3).**
*Low-carbon city policies are more conducive to green technology innovation in high-carbon emission industries than in low-carbon emission industries.*


### 2.4. Low-Carbon City Pilot Policies and Financing Constraints

Innovation in the field of green technology has the characteristics of high uncertainty, large capital needs, and strong externalities. Many companies, especially listed ones, often face great financing constraints. Under the pressure of profitability, these firms pay more attention to short-term benefits, focus on their current financial statements, and lack the enthusiasm for long-term investment, which inhibit them from developing independent R&D and collaborative innovation [37,38]. To ease the constraints in accumulating scientific research funds for enterprises and optimizing the allocation of resources among industries, low-carbon pilot cities have introduced green finance policies and provided financial support through tax incentives, government subsidies, and special funds. Many studies have pointed out that government subsidies can alleviate financing constraints [39,40] and support enterprises in their pursuit of green innovation activities [41], tax incentives reduce the financial pressure faced by these enterprises to a certain extent, promote their scientific and technological research, and improve their technology level [42,43,44]. However, some scholars believe that government subsidies may squeeze out enterprise R&D investment [45], and the subsequently triggered rent-seeking activities will reduce the incentive effect of these subsidies [46]. From the perspective of government subsidies and tax incentives, this study examines whether financing constraints can be eased to promote the implementation of low-carbon pilot policies to promote corporate green technology innovation while providing data support for future low-carbon pilot work. Based on the above discussion, Hypothesis 4 is proposed as follows.

**Hypothesis** **4** **(H4).**
*To implement a low-carbon city pilot policy, enterprises should promote green technological innovation by alleviating financing constraints.*


## 3. Methodology and Data

### 3.1. Data Sources

Given the availability, accuracy, and relative completeness of listed company data, we used Chinese A-share listed companies as our sample and selected the green patent data of listed companies from 2005 to 2019, and the corresponding economic data at the company, industry, and city levels for an empirical analysis. At the company level, the patent data were collected from the State Intellectual Property Office of the People’s Republic of China; other data were collected from the Guotaian database. At the industry level, data were obtained from the China Statistical Yearbook, and the carbon intensity of the industry was measured after processing these data. At the city level, these data were obtained from the China City Statistics Yearbook. Given that innovation in the field of green technology is mainly concentrated in the industrial sector, we mainly focused on the secondary industry and excluded the financial services and the agriculture, forestry, animal husbandry, and fishery sectors. We also excluded those enterprises with incomplete and abnormal data (including listed companies under special treatment).

Our analysis of low-carbon pilot policies covered three batches of low-carbon pilot provinces and cities (the Daxing’anling area was not included due to missing data). Given that, specific pilot work is mainly promoted in cities. Accordingly, we subdivided low-carbon pilot provinces into prefecture-level cities and examined the overlapping areas between batches. For example, Shenyang, Zhongshan, and other cities were included in the third batch of pilot cities, but their province was included in the first batch of pilot provinces. Therefore, these cities were included in the first batch instead of the third batch. The low-carbon pilot areas examined in this research covered 102 prefecture-level cities. Given that the innovation in the field of green technology often requires a long period, the promotion effect of the pilot policy on the green technology innovation of enterprises may be relatively limited in the initial stage. Moreover, the scope of low-carbon pilots was re-expanded in 2012; the interval between the two pilots was very short. By combining the actual situation with the previous research [47], we took 2012 and 2017, when the scope of the pilot was expanded, as the time node of the pilot policy, and analyzed the promotion effect of the low-carbon city pilot policy on the green technological innovation of enterprises.

### 3.2. Variable Description

#### 3.2.1. Dependent Variable

We took the number of green patent applications (lpt) of listed companies, including green invention (lpt1) and green utility (lpt2) patents, as our explained variable. On the one hand, patent technology has a long application period and may start to affect enterprise performance during the process. Therefore, the number of patent applications is highly reliable and timely [47]. On the other hand, the number of green patent applications is highly intuitive and shows that the investment of enterprises in this area is clearer than the relatively general scientific research investment. Moreover, the data of green patent applications can be classified to reflect the value connotation of innovation activities.

#### 3.2.2. Core Explanatory Variable

We set a dummy variable as our core explanatory variable that takes a value of 1 if the company’s city will be included in the low-carbon city pilot list after 2012 or 2017, and takes a value of 0 otherwise.

#### 3.2.3. Control Variable

We chose other factors that may affect the green technological innovation of companies at the enterprise, industry, and city levels as our control variables. (1) Enterprise maturity (lnage). The longer a company goes public, the more mature its prospect planning, goal setting, and development focus become. The company also becomes highly sensitive to policy changes and develops a strong sense of innovation. We measured this indicator by the logarithm of the number of years that the company has been listed (to avoid the influence of the number of listing years being 0, we increased the number of listing years by 1 and took the logarithm). (2) Enterprise size (lnsize). The scale of an enterprise is closely related to its technological innovation [48]. We used the logarithm of total capital at the end of the year to measure this indicator. (3) Corporate debt (lndebt). The debt situation of enterprises reflects the evaluation of the market on the credit ability of enterprises [49]. We used the logarithm of the ratio of a company’s total liabilities to its total assets at the end of the period to measure this indicator. (4) TobinQ (lnTobinQ). A larger TobinQ value corresponds to more social wealth created by an enterprise and a stronger sense of innovation. We dealt with this indicator by using a logarithm. (5) The related variables of corporate performance. Given that corporate performance and capital structure affect corporate innovation in green technology, we took corporate return on total assets (ROA), capital intensity (LNCaP), and number of employees (lnlabor) as control variables. We measured ROA by the proportion of a company’s net profit in its total assets, LNCaP by the logarithm of the ratio of a company’s total assets to its operating income, and lnlabor by the logarithm of the number of employees in a year. (6) City-level variables. Given that the degree of openness of cities, their industrial structure, economic status, and other environmental regulations will affect the performance of enterprises in green technology innovation [32], we controlled the use of foreign capital (lnfdigdp), industrial structure (ind), gross domestic product (GDP) per capita (lnpergdp), and other environmental regulations (lnso_2_). We measured lnfdigdp by the logarithm of the actual use of foreign capital to regional GDP, ind by the proportion of the secondary industry, lnpergdp by the logarithm of regional per capita GDP, and lnso_2_ by the logarithm of the proportion of regional sulfur dioxide emissions to regional GDP. Descriptive statistics of main variables are presented in Table 1.

### 3.3. Model Settings

Referring to Cheng [23] and Song [26], we used a multi-period DID model to explore the impact of low-carbon city pilot policies on corporate green technology innovation. The low-carbon pilot areas were included in the treatment group, whereas the non-pilot areas were included in the control group. The model was constructed as
(1)$$lpt_{it}=\alpha_{0}+\alpha_{1}lcpilot_{r}\times post_{rt}+\alpha_{2}\sum^{\;}control_{it}+\gamma_{t}+\mu_{j}+\lambda_{r}+\xi_{it}$$
where *lpt_it_* is the number of green patent applications of a listed company in year t, *lcpilot* is the dummy variable of a low-carbon city that equals to 1 if the company’s registration place is a green low-carbon pilot city and equals to 0 otherwise, and *post_rt_* is a dummy variable of the time when the pilot policy was issued (given that the notices for the establishment of the second batch of pilot cities were announced on 26 November, 2012, the supporting policies could not be easily implemented in the pilot areas within the year. We set the first two batches of pilot policies to take effect in 2013 and the third batch of pilot policies to take effect in 2017). If the company is included among the first two batches of pilot areas announced after 2012, then *post_rt_* takes a value of 1. Otherwise, this variable takes a value of 0. If this company is included among the third batch of pilot areas in or after 2017, then *post_rt_* takes a value of 1 and a value of 0 otherwise. In addition, *Control* represents the control variable, *γ_t_*, *μ_j_*, and *λ_r_* control the fixed effects of time, industry, and city, respectively, *ξ_it_* is a random interference item, *i*, *j*, and *r* represent an enterprise, industry, and city, respectively, and *t* represents time. In this model, the empirical analysis examined whether the coefficient of the double difference term *lcpilot_r_* × *post_rt_* was positively significant. If *α_1_* is significantly greater than 0, then the low-carbon pilot policy effectively promotes the innovation of enterprises in the field of green technology.

## 4. Empirical Results

We initially performed a benchmark regression to analyze the impact of low-carbon city pilot policies on corporate green technological innovation, to test whether this policy has a promoting effect, and to conduct robustness tests based on the results.

### 4.1. Benchmark Estimation Results

This section quantitatively analyzes the impact of introducing low-carbon city pilot policies on the green technology innovation of A-share listed companies based on the benchmark model. The results are shown in Table 2, where columns (1) and (2) show the impact of the pilot policy on the number of green technology patent applications, columns (3) and (4) show the number of green invention patents, and columns (5) and (6) show the impact of the policy on green utility patents. The fixed effects of time, industry, and city were added in columns (2), (4), and (6).

The regression results in Table 2 reveal that the coefficient of lcpliot × post is significantly positive and remains significant at the 1% level after controlling for the fixed effect, thereby suggesting that the pilot policy of low-carbon cities effectively promotes the innovation of enterprises in green technology. After the introduction of the pilot policy, the number of green patent applications of listed enterprises in the industrial sector significantly increased, especially for green invention patents, thereby supporting Hypothesis 1, which posits that the pilot low-carbon city policy is conducive to enterprise green technology innovation. On the one hand, given that the pilot policy focuses on the industry, construction, and other sectors, under the requirements of low-carbon development, the emission cost of relevant enterprises increases, their energy conservation and emission reduction incentives gradually increase, and their enthusiasm for green technology innovation is improved. On the other hand, given the weak constraint of the pilot policy, the pilot cities can flexibly promote their work according to their actual situation and explore the win–win path of emission reduction and economic growth. The cost pressures faced by enterprises related to pollution control and emission reduction will be controlled within a relatively reasonable range to avoid crowding out investments in R&D and innovation. In terms of control variables, the increase in debt, scale, ratio of market value to replacement value, capital intensity, number of employees, use of urban foreign capital, and industrial structure all triggered an increase in green patent applications to a certain extent, which was consistent with our expectations during our selection of variables. However, enterprise maturity showed a negative impact on green technology innovation. Specifically, the longer an enterprise goes public, the weaker its innovation enthusiasm becomes. The investment behavior of an enterprise may be constrained by financing capital after going public, which may drive this enterprise to focus on its current profit and enhance its dependence on its current development path.

### 4.2. Parallel Trend Test

In the above empirical analysis, we preliminarily evaluated the promotion effect of low-carbon city pilot policies on corporate green technological innovation. Given that meeting the parallel trends assumption is an important prerequisite for the use of multi-period DID, we followed the practice of Song [26] and constructed our model as
(2)$$lpt_{it}=\alpha_{0}+\overset{5}{\underset{k=-5}{\sum }}\beta_{k}\times Z_{r,t_{0}+k}+\alpha_{1}\sum^{\;}control_{it}+\gamma_{t}+\mu_{j}+\lambda_{r}+\xi_{it}$$
where *t*_0_ represents the time when the low-carbon pilot policy was introduced, K represents the k-th year after the policy was introduced, and *Z_r__,_**_t_*0*__*_*+k*_ are dummy variables indicating the k-th year after the city *r* was included in the low-carbon city pilot scope (the value range of K is [−5,5], which covers the 5 years before and after the policy, was issued. If the time exceeds five years, then it is still set to five years in the model). Meanwhile, the estimated coefficient *β_K_* reflects the difference in the number of green patent applications between the processing and control groups K years after the introduction of the pilot policy. When k is less than 0 (before the introduction of the policy) if the trend of *β_K_* is stable and fluctuates around 0, then the parallel trend hypothesis is satisfied. Otherwise, a significant difference is observed between the two groups before the introduction of the pilot policy, thereby rejecting our hypothesis. The test results are shown in Figure 1. We took the previous period of the policy as the benchmark group, the vertical axis as the dynamic effect of the policy (represented by the value of *β_K_*), and the horizontal axis as the time of the policy. Figure 1 shows that before the introduction of the pilot policy for low-carbon cities, *β_K_* fluctuates around 0, thereby indicating no significant differences between the treatment and control groups before the introduction of the policy. This finding is comparable and conforms to the parallel trend hypothesis. Specifically, in the year when the policy was issued and several years after, *β_K_* remained positive and significant. Despite the large fluctuation in the second and third years, the overall trend was still on the rise, thereby suggesting that the pilot policy of low-carbon cities increased the number of green patent applications of enterprises and that such promotion effect was gradually enhanced.

### 4.3. Robustness Test

#### 4.3.1. PSM–DID

To avoid the sample self-selection problem caused by systematic differences, we adopted the double differential propensity score matching method (PSM–DID) for the robustness test. According to the balance test results in Table 3, the standardized deviation of most covariates after nearest neighbor matching was less than 10%, and the original hypothesis that there is no systematic deviation between the treatment and control groups was supported. On this basis, we performed a multi-period double difference estimation and reported the results in Table 4. The coefficient of lcpilot × post remained positive and significant, thereby suggesting that through the robustness test, the pilot policy of low-carbon cities can promote the green technology innovation of enterprises. This result was also in line with the empirical findings of the benchmark regression.

#### 4.3.2. Replace Dependent Variables

To eliminate other unobservable factors that interfere with the conclusions of the regression model, we used the index of the proportion of green patent applications in all patent applications in a specific year to test the robustness [50], to reflect the importance of enterprises in green technology innovation, and to further highlight the impact of the low-carbon city pilot policy. The regression results are shown in Table 5, where *ralpt*, *ralpt1*, and *ralpt2* denote the proportion of green patents, green invention patents, and green utility patents, respectively. According to the regression results, the coefficients of *lcpilot × post* were positively significant, suggesting that the introduction of the low-carbon city pilot policy promotes the green technology innovation of enterprises, improves their attention, and proves the robustness of the benchmark regression results. Moreover, the partial coefficient of double difference in column (4) was significantly larger than that in column (6), thereby suggesting that the promotion effect of the low-carbon city pilot policy on enterprise green technology innovation is mainly reflected in green invention patents with high scientific and technological content.

### 4.4. Heterogeneity Analysis

#### 4.4.1. Heterogeneity Test Based on Region

Chinese regions show huge differences in their economic conditions, resource endowments, and business environments. Although the inherent location and environmental deficiencies of the central and western regions have been filled through the continuous advancement of infrastructure construction and the forging of new fulcrums for development, a considerable gap can still be observed among the eastern, central, and western cities, which will influence their investment activities, such as enterprise R&D and innovation. Accordingly, we examined the geographical distribution of enterprises and divided our sample into eastern, central, and western cities based on the division of national policies. Following Wang [28] and Xiong [34], we introduce regional dummy variables into model (1) to explore whether low-carbon city pilot policies have a heterogeneous impact on the green technology innovation of enterprises across different regions. The model was constructed as
(3)$$lpt_{it}=\beta_{0}+\beta_{1}lcpilot_{r}\times post_{rt}\times location+\beta_{2}\sum^{\;}control_{it}+\gamma_{t}+\mu_{j}+\lambda_{r}+\xi_{it}$$
where *location* represents the location of the city where the company belongs (i.e., east, middle, and west). While examining the effect of low-carbon pilot policies on the green technological innovation of enterprises in eastern cities, we set east, middle, and west to 1, 0, and 0, respectively; similarly, while investigating central cities (western cities), we set middle (west) to 1, and the rest was set to 0.

The regression results are shown in Table 6. The coefficients in columns (1) and (4) were positively significant at the 5% level, thereby indicating that with the gradual implementation of the low-carbon pilot policy, enterprises in eastern cities begin to focus on green technology innovation, which increases not only the number, but also the proportion of green patent applications. However, the coefficients in columns (2) and (3) were not significant, indicating that the promotion effect of the low-carbon pilot policy on the green technology innovation of enterprises in central and western cities was not obvious. Although significant, the coefficient in column (6) only reflected the relative level of green innovation, thereby supporting Hypothesis 2. Specifically, the pilot policy of low-carbon cities was more conducive to the enterprise green technology innovation of eastern cities than to those of the central and western cities. On the one hand, because of the objective differences in the economic structure of the eastern, central, and western cities, traditional industries are mainly concentrated in the central and western cities, limited by their regional economy and traditional development path, enterprises in central and western cities face more difficulties in their attempts to transform and upgrade themselves and have a relatively weak innovation enthusiasm compared with their counterparts in eastern cities. On the other hand, some differences can be observed in the governance capacity of regional governments. The ability of some pilot cities to explore regional development mode needs to be improved, and they cannot utilize the low-carbon pilot policy to guide enterprises to explore green technology innovation.

#### 4.4.2. Heterogeneity Test Based on Industry Carbon Emissions

The energy conservation, emission reduction, and green technology innovation of high carbon emission industries play key roles in implementing low-carbon pilot work and promoting the development of a low-carbon economy. As an important focus of carbon emission control, high carbon emission industries are often subject to many constraints, and relevant enterprises develop a strong response to the pilot policy of low-carbon cities and actively increase their R&D investments to promote innovation in the field of green technology. To test this hypothesis and further explore the effectiveness of the slow-carbon city pilot policy, we introduced industry carbon intensity (*co*_2_) into model (1) and constructed a triple difference model as shown in model (4). Industry carbon intensity denotes the ratio of industry carbon emissions and operating income. The carbon emission coefficient refers to the “General Principles of Comprehensive Energy Consumption Calculation” (GB/T 2589-2008) and “Provincial Greenhouse Gas Inventory Compilation Guide” (Fagaiban Climate (2011) No. 1041). We focused on the coefficient of the triple difference sub item *lcpilot* × *post* × *co*_2_. If this coefficient is less than 0, then the pilot policy has a stronger role in promoting green technology innovation in low-carbon emission industries than in high-carbon emission industries. Otherwise, high-carbon emission industries are the main objects of the policy-induced green technology innovation.
(4)$$\begin{array}{c} \;lpt_{it}=\beta_{0}+\beta_{1}lcpilot_{r}\times post_{rt}\times co_{2}+\beta_{2}lcpilot_{r}\times post_{rt}+\beta_{3}lcpilot_{r}\times co_{2}\\ +\beta_{4}post_{rt}\times co_{2}+\beta_{5}\sum^{\;}control_{it}+\gamma_{t}+\mu_{j}+\lambda_{r}+\xi_{it}\; \end{array}$$

The regression results are shown in Table 7. The *lcpilot* × *post* × *co*_2_ coefficients were significantly positive at the 1% level, thereby suggesting that the pilot policy plays a key role in promoting green technology innovation in high carbon emission industries. To achieve the peak goal of carbon emissions and control the amount of carbon emissions, the local government took the regulation of high carbon emission industries as its starting point and encouraged the relevant enterprises to increase their investment in green technology R&D to meet social development needs. This finding echoed those of Xu [47] and Sheng [51], and verified Hypothesis 3.

### 4.5. Mechanism Analysis

The above empirical test results show that the pilot policy of low-carbon cities can promote the green technology innovation of enterprises, especially those in eastern cities, which belong to high carbon emission industries. To discuss the effectiveness of this policy, we further analyzed its mechanism for easing financing constraints. Compared with traditional technology innovation, green technology innovation focuses on protecting the ecological environment, and coordinating and unifying economic and ecological benefits. However, green technology R&D has high uncertainty and positive externality, while enterprises have relatively low enthusiasm and face huge constraints in accumulating R&D funds. To implement low-carbon pilot work, the government requires financial support. We therefore analyzed the mechanism of the low-carbon city pilot policy from the perspective of easing financing constraints and tested whether such policy can promote the innovation of enterprises in green technology through tax incentives and government subsidies. The triple difference model is shown in models (5) and (6), where *tax* denotes tax incentives, whereas *lnsub* denotes government subsidies (tax is computed as “tax refund received/(tax refund received + various taxes paid),” and lnsub is the logarithm of government subsidies. The data were collected from the Guotaian database).
(5)$$\begin{array}{c} lpt_{it}=\beta_{0}+\beta_{1}lcpilot_{r}\times post_{rt}\times tax+\beta_{2}lcpilot_{r}\times post_{rt}+\beta_{3}lcpilot_{r}\times \\ tax+\beta_{4}post_{rt}\times tax+\beta_{5}\sum^{\;}control_{it}+\gamma_{t}+\mu_{j}+\lambda_{r}+\xi_{it} \end{array}$$
(6)$$\begin{array}{c} lpt_{it}=\beta_{0}+\beta_{1}lcpilot_{r}\times post_{rt}\times lnsub+\beta_{2}lcpilot_{r}\times post_{rt}+\beta_{3}lcpilot_{r}\times lnsub\\ +\beta_{4}post_{rt}\times lnsub+\beta_{5}\sum^{\;}control_{it}+\gamma_{t}+\mu_{j}+\lambda_{r}+\xi_{it} \end{array}$$

The empirical results are shown in Table 8. The triple difference partial coefficients in columns (1) to (4) were positively significant, whereas those in columns (5) and (6) were not significant. These results suggest that the pilot policy of low-carbon cities alleviates the financing constraints of enterprises through tax incentives and government subsidies and encourages them to carry out green technology innovation as reflected in their application of green invention patents. Although the promotion effect on their application of green practical patents is relatively not obvious, it does not affect the verification of Hypothesis 4, especially considering that the innovation level of green invention patents is higher than that of utility patents. Thereby supporting Hypothesis 4. Columns (1) to (4) also show that the size and significance level of the partial coefficients of triple difference in model (5) are significantly higher than that in model (6), which may be ascribed to the fact that tax preference plays an effective role through an open and transparent market mechanism while reducing direct intervention in enterprises. In this case, enterprises can give full play to their subjective ability to optimize their allocation of resources based on their current situation. As a direct incentive measure, government subsidies often have strict control, which hinders enterprises from making adjustments according to their actual situation. In addition, given the strong planning of government subsidies and limited by incomplete information, the ability of competent departments, and the intervention of interest groups, some problems may emerge, such as blind subsidies for enterprises with poor performance and blind construction of high-tech projects.

## 5. Discussion

To cope with climate change and achieve sustainable development, green and low-carbon economy has become a new focus in China. However, the industrial sector still dominates the industrial structure of most cities in China, and this sector’s energy structure is mainly concentrated in high-carbon energy. Many challenges are involved in the process of low-carbon development, which necessitate the market to play its key role in stimulating the green innovation vitality of enterprises. We constructed a multi-period double difference model to study the impact of the low-carbon city pilot policy on the green technology innovation of enterprises. Our research contributes to the literature in two important aspects. First, we expand the relevant literature on the low-carbon city pilot policy. Previous studies have often focused on the industry or city level [22,23,24,25,26,27,28] and tested industrial structure [23,24,25], carbon emissions [22,27], and other indicators yet largely ignored the impact of low-carbon city construction from the perspective of enterprises. Although some scholars [34,47] have examined this aspect in recent years, most of the sample data included only the first two batches of pilot cities. We used multi-phase DID to include three batches of pilot cities within our empirical scope and found that the pilot policy of low-carbon cities promotes the green technology innovation of enterprises and increases the number and proportion of their green patent applications, which may be ascribed to the emphasis of this policy on industrial sectors and its weak constraints. Enterprises are encouraged by innovation and avoid crowding out their R&D investments due to high pollution control cost. Second, we clarified the role and impact mechanism of low-carbon pilot policy, and found that the introduction of the pilot policy was highly conducive to the green technology innovation of enterprises in eastern cities and high carbon emission industries. By further analyzing the mechanism of the pilot policy in easing financing constraints, we found that tax incentives and government subsidies help this policy play its role in green technology innovation. Moreover, tax preference has a stronger promotion effect compared with government subsidies probably because the former exerts its effect through the market mechanism. In addition, enterprises have a strong subjective initiative and can be adjust themselves in time according to their actual situation. With its strong planning, government subsidies may blindly subsidize enterprises with poor performance and blindly construct high-tech projects.

We preliminarily analyzed the role of pilot policies in low-carbon cities in promoting the green technology innovation of enterprises. However, given that pilot policies themselves are still in the process of promotion, some limitations need to be monitored and analyzed. We used a multi-period double difference method to test the green technology innovation effect of the pilot policy. However, the third batch of the low-carbon city pilot work was carried out within a short period. In this case, considering the policy lag, corporate data covering a longer sample period are required for subsequent verification. In our analysis of the mechanism for alleviating financing constraints, our selected government subsidy indicators were not subdivided into scientific and technological innovation projects, could not fully reflect the incentive effects, and require further testing by mining relevant data. The financial constraints and investment risks of enterprises in green technological innovation are issues that need to be solved urgently in the development of low-carbon cities. Follow-up studies can start from the circular business model [52,53], analyze the effectiveness of circular economy in reducing resource consumption, waste and emissions, explore the feasibility of building a circular economy development model in high-carbon industries, and expand relevant research on low-carbon city pilot policies.

## 6. Conclusions and Suggestions

Based on a sample of A-share listed companies from 2005 to 2019, we analyzed the impact of the low-carbon city pilot policy on the green technology innovation of enterprises by using a multi-period double difference model. Based on the model results, we conducted a heterogeneity analysis, explored the direction of the pilot policy, and analyzed its impact mechanism to provide empirical basis for the follow-up implementation and promotion of this pilot policy. We obtained three key findings. First, the pilot policy of the low-carbon city promoted the innovation of enterprises in the field of green technology and significantly increased their number and proportion of green patent applications, especially green invention patents. Second, our heterogeneity analysis revealed that compared with central and western cities, the pilot policy was more conducive to the green technology innovation of enterprises in eastern cities. Meanwhile, compared with low-carbon emission industries, the pilot policy had a bigger role in promoting green innovation in high-carbon emission industries. Third, our mechanism analysis identified tax incentives and government subsidies as important fiscal and tax tools for fulfilling the roles of the low-carbon city pilot policy. By easing their financing constraints, enterprises could effectively promote green technology innovation. Tax incentives play an important role in promoting the green technology innovation of enterprises and mainly focus on green invention patents.

To further promote the development of a low-carbon economy and achieve a comprehensive green transformation of economic and social development, we propose the following countermeasures and suggestions: (1) actively promote the implementation and diffusion of pilot policies of low-carbon cities. The low-carbon pilot policy can promote the green technology innovation of enterprises, which is in line with China’s current development situation and meets the needs of social development. To develop a low-carbon economy, we should actively promote the pilot work. On the one hand, we should sort out the achievements and shortcomings of the three batches of low-carbon city pilot work, summarize the work experience, and form typical cases to promote. On the other hand, according to their regional economic structure and recent development, we should gradually expand the scope of pilot cities and pursue carbon peak and carbon neutralization. (2) Carry out the pilot work of low-carbon cities according to local conditions. Based on the objective conditions of regions and industries, we should scientifically formulate an implementation plan of the pilot work to effectively control the amount and intensity of carbon emissions and gradually establish a pilot evaluation and retirement mechanism to supervise and restrict low-carbon pilot cities. (3) Attach importance to guiding the transformation and upgrading of high-carbon industries. Promoting the low-carbon development of high-carbon industries is an important task in the low-carbon pilot work. The pilot government can actively strive to levy carbon tax in the form of local tax. Through the collection of carbon tax and the establishment of green development fund, the government can guide and encourage high-carbon industries to promote the technological transformation and upgrading of traditional industries, and carry out green technology innovation, so as to give full play to the market main role of enterprises in developing low-carbon industries and developing clean technologies. At the same time, the green monitoring and evaluation system needs to be constructed, and the information of enterprise environmental violations should be included in the credit information. This will provide the basis for the government to implement fiscal and tax policies, so that the government can give full play to the role of tax incentives and government subsidies, ease the financing constraints of related enterprises, promote green technology innovation, and gradually optimize the regional industrial structure.

## Figures and Tables

**Figure 1 ijerph-18-03695-f001:**
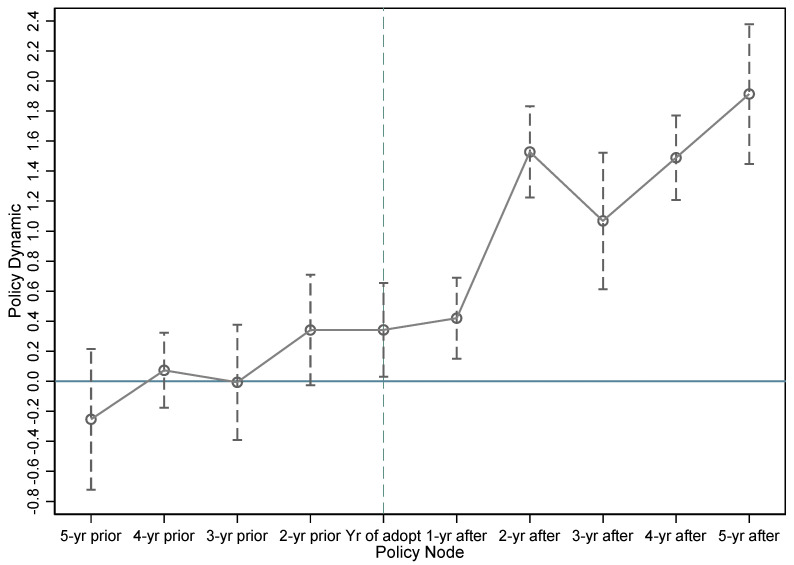
Parallel trend test results.

**Table 1 ijerph-18-03695-t001:** Summary statistics.

Variables	Variable Definition	Mean	SD	Min	Max
lpt	Green patent applications	1.538	11.9415	0.0000	626.0000
lpt1	Green invention patent applications	0.9676	8.3685	0.0000	450.0000
lpt2	Green utility patent applications	0.5709	4.6334	0.0000	195.0000
lcpilot	Dummy variable of a low-carbon city	0.6751	0.4683	0.0000	1.0000
post	Dummy variable of the time	0.2714	0.4447	0.0000	1.0000
lnage	Logarithm of listing age	2.5729	0.4787	0.0000	3.4012
lnsize	Logarithm of total corporate assets	22.2212	1.3738	16.5083	27.4673
lndebt	Logarithm of corporate debt ratio	−0.7659	0.5442	−4.9510	4.5743
lnTobinQ	Logarithm of TobinQ	0.5113	0.5074	−1.8786	4.2436
ROA	Return on total assets of the enterprise	0.0326	0.2743	−8.7534	22.0051
LNCaP	Logarithm of capital intensity	0.5484	0.7133	−1.7633	13.5776
lnlabour	Logarithm of the number of employees	7.9903	1.3070	1.0986	12.2900
lnfdigdp	The logarithm of the ratio of actually used foreign capital to GDP	−5.7097	1.0456	−15.5597	−3.5101
ind	Proportion of secondary industry to GDP	0.4634	0.1081	0.1814	0.8592
lnpergdp	Logarithm of GDP per capita	10.8439	0.7440	7.8474	13.0557
lnso_2_	The logarithm of the ratio of regional sulfur dioxide emissions to GDP	0.4308	1.7227	−5.5121	5.6520

**Table 2 ijerph-18-03695-t002:** Impact of low-carbon city pilot policies on corporate green patent applications.

Variables	Lpt		Lpt1		Lpt2	
	(1)	(2)	(1)	(2)	(1)	(2)
lcpliot × post	0.881 ***	0.679 ***	0.511 ***	0.449 ***	0.370 ***	0.229 **
	(0.171)	(0.163)	(0.149)	(0.007)	(0.101)	(0.083)
lndebt	0.239 *	0.329 ***	0.220 ***	0.282 ***	0.018	0.047 *
	(0.129)	(0.165)	(0.071)	(0.053)	(0.075)	(0.150)
lnsize	1.608 ***	0.564 **	1.000 **	0.420 **	0.608 ***	0.144 ***
	(0.519)	(0.067)	(0.356)	(0.043)	(0.176)	(0.030)
ROA	−0.142	0.078	−0.063	0.096	−0.080	−0.018
	(0.232)	(0.110)	(0.146)	(0.055)	(0.095)	(0.056)
lnTobinQ	1.682 *	0.340	1.056 *	0.256	0.626 **	0.084
	(0.793)	(0.196)	(0.539)	(0.160)	(0.268)	(0.056)
LNCaP	0.087	0.421 ***	0.011	0.233 **	0.077	0.188 ***
	(0.110)	(0.109)	(0.085)	(0.096)	(0.079)	(0.032)
lnlabour	0.599 **	0.434 ***	0.452 **	0.322 ***	0.147 **	0.112 ***
	(0.224)	(0.099)	(0.177)	(0.077)	(0.050)	(0.028)
lnage	−0.684 *	−0.878 ***	−0.551 *	−0.452 ***	−0.133	−0.426 ***
	(0.356)	(0.129)	(0.301)	(0.093)	(0.077)	(0.081)
ind	2.345	2.753 ***	1.880	2.255 ***	0.465	0.498 ***
	(1.785)	(0.820)	(1.249)	(0.615)	(0.573)	(0.347)
lnfdigdp	0.353 ***	0.143 ***	0.199 **	0.101 **	0.154 ***	0.042 ***
	(0.116)	(0.007)	(0.086)	(0.006)	(0.042)	(0.002)
lnpergdp	−0.178	−1.052	−0.118	−0.644	−0.060	−0.408
	(0.271)	(0.164)	(0.168)	(0.159)	(0.130)	(0.050)
lnso_2_	−0.114	−0.060	−0.199	−0.100 *	0.086	0.040
	(0.187)	(0.078)	(0.137)	(0.051)	(0.061)	(0.031)
constant	−35.278 ***	−3.514 **	−22.337 **	−4.377 **	−12.941 ***	0.866
	(10.679)	(0.992)	(7.674)	(1.515)	(3.116)	(1.008)
Time fixed effect	NO	YES	NO	YES	NO	YES
Industry fixed effect	NO	YES	NO	YES	NO	YES
Urban fixed effect	NO	YES	NO	YES	NO	YES
N	10,080	10,080	10,080	10,080	10,080	10,080
Adj R2	0.050	0.032	0.047	0.029	0.037	0.026

Note: figures in () are robust standard error; ***, ** and * indicate significance at the 1%, 5%, and 10% levels, respectively.

**Table 3 ijerph-18-03695-t003:** Propensity score matching method–difference-in-differences (PSM–DID) balance test results.

Variables		Mean		SD/%	T-Test	
		Treat	Control		T	*p* > |t|
lndebt	U	−0.8177	−0.7483	−12.9	−5.76	0.000
	M	−0.8025	−0.8199	3.2	1.09	0.278
lnage	U	2.9128	2.4449	123.2	28.10	0.000
	M	2.8865	2.8920	−1.5	−0.79	0.432
ROA	U	0.0373	0.0302	3.0	1.15	0.250
	M	0.0367	0.0585	−9.2	−1.53	0.125
lnsize	U	22.8380	22.0160	61.9	27.76	0.000
	M	22.7490	22.6080	10.6	1.62	0.112
lnTobinQ	U	0.5221	0.5056	3.3	1.45	0.146
	M	0.5383	0.5182	4.0	1.31	0.190
lnfdigdp	U	−5.5682	−5.7659	19.5	8.42	0.000
	M	−5.6442	−6.1207	46.9	10.91	0.000
lnpergdp	U	11.4470	10.6250	136.9	56.47	0.000
	M	11.3220	11.2290	15.6	6.34	0.000
LNCaP	U	0.5834	0.5283	7.9	3.48	0.001
	M	0.5770	0.5661	1.6	0.53	0.595
lnlabour	U	8.2002	7.9290	20.9	9.31	0.000
	M	8.1869	8.1295	4.4	1.50	0.134
ind	U	0.3969	0.4883	−87.9	−40.53	0.000
	M	0.4264	0.4299	−3.4	−1.08	0.281
lnso_2_	U	−1.2446	1.0463	−152.4	−73.26	0.000
	M	−0.5973	−0.5601	−2.5	−0.99	0.324

Note: U denotes before matching, whereas M denotes after matching.

**Table 4 ijerph-18-03695-t004:** PSM–DID robustness test.

Variables	Lpt		Lpt1		Lpt2	
	(1)	(2)	(3)	(4)	(5)	(6)
lcpilot × post	0.470 ***	0.683 ***	0.253 ***	0.410 ***	0.217 **	0.273 **
	(0.137)	(0.145)	(0.068)	(0.069)	(0.079)	(0.091)
Control	NO	YES	NO	YES	NO	YES
Time fixed effect	YES	YES	YES	YES	YES	YES
Industry fixed effect	YES	YES	YES	YES	YES	YES
Urban fixed effect	YES	YES	YES	YES	YES	YES
N	7625	7625	7625	7625	7625	7625
Adj R2	0.087	0.126	0.086	0.124	0.082	0.116

Note: figures in () are robust standard error; ***, ** indicate significance at the 1%, 5%levels, respectively.

**Table 5 ijerph-18-03695-t005:** Impact of low-carbon city pilot policies on the proportion of green patent applications.

Variables	Ralpt		Ralpt1		Ralpt2	
	(1)	(2)	(3)	(4)	(5)	(6)
lcpilot × post	0.006 ***	0.006 ***	0.006 ***	0.005 ***	0.002 **	0.001 **
	(0.003)	(0.002)	(0.002)	(0.001)	(0.001)	(0.000)
Control	YES	YES	YES	YES	YES	YES
Time fixed effect	NO	YES	NO	YES	NO	YES
Industry fixed effect	NO	YES	NO	YES	NO	YES
Urban fixed effect	NO	YES	NO	YES	NO	YES
N	10,080	10,080	10,080	10,080	10,080	10,080
Adj R2	0.025	0.035	0.027	0.036	0.024	0.032

Note: figures in () are robust standard error; ***, ** indicate significance at the 1%, 5% levels, respectively.

**Table 6 ijerph-18-03695-t006:** Heterogeneity analysis of region.

Variables	Lpt			Ralpt		
	(1)	(2)	(3)	(4)	(5)	(6)
lcpilot × post × east	0.125 **			0.004 **		
	(0.050)			(0.001)		
lcpilot × post × middle		0.012			0.010	
		(0.075)			(0.005)	
lcpilot × post × west			0.019			0.012 ***
			(0.047)			(0.003)
Control	YES	YES	YES	YES	YES	YES
N	10,080	10,080	10,080	10,080	10,080	10,080
AdjR2	0.053	0.053	0.053	0.016	0.016	0.017

Note: figures in () are robust standard error; ***, ** indicate significance at the 1%, 5% levels, respectively.

**Table 7 ijerph-18-03695-t007:** Heterogeneity analysis of industry carbon emissions.

Variables	Lpt	Lpt1	Lpt2	Ralpt	Ralpt1	Ralpt2
	(1)	(2)	(3)	(4)	(5)	(6)
lcpilot × post × co_2_	0.763 ***	0.348 ***	0.295 ***	0.022 ***	0.010 ***	0.012 ***
	(0.204)	(0.074)	(0.581)	(0.002)	(0.002)	(0.001)
lcpilot × post	0.042	0.006	0.037	0.001	0.001	0.000
	(0.032)	(0.017)	(0.063)	(0.002)	(0.001)	(0.000)
Control	YES	YES	YES	YES	YES	YES
Time fixed effect	YES	YES	YES	YES	YES	YES
Industry fixed effect	YES	YES	YES	YES	YES	YES
Urban fixed effect	YES	YES	YES	YES	YES	YES
N	9886	9886	9886	9886	9886	9886
Adj R2	0.054	0.045	0.018	0.024	0.019	0.023

Note: figures in () are robust standard error; ***indicate significance at the 1%levels, respectively.

**Table 8 ijerph-18-03695-t008:** Mechanism analysis of the low carbon city pilot policy.

Variables	Lpt		Lpt1		Lpt2	
	(1)	(2)	(3)	(4)	(5)	(6)
lcpilot × post	−0.148	−0.139	−0.341 ***	−0.310	0.192 ***	0.028
	(0.120)	(0.166)	(0.073)	(0.315)	(0.064)	(0.028)
lcpilot × post × tax	6.428 ***		6.157 ***		0.270	
	(1.019)		(0.812)		(0.239)	
lcpilot × post × lnsub		0.017*		0.051 *		0.002
		(0.009)		(0.025)		(0.002)
Control	YES	YES	YES	YES	YES	YES
Time fixed effect	YES	YES	YES	YES	YES	YES
Industry fixed effect	YES	YES	YES	YES	YES	YES
Urban fixed effect	YES	YES	YES	YES	YES	YES
N	9904	9904	9904	9904	9904	9904
AdjR2	0.024	0.022	0.025	0.020	0.016	0.016

Note: Figures in () are robust standard error; ***, * indicate significance at the 1%, 10% levels, respectively.

## Data Availability

Not applicable. No new data were created or analyzed in this study.

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
