# Peer review of "Does the Low-Carbon City Pilot Policy Promote Green Technology Innovation? Based on Green Patent Data of Chinese A-Share Listed Companies"

_ijerph, 2021, doi:10.3390/ijerph18073695_

Round 1

Reviewer 1 Report

The article presents the results of intense work and the study is one of topicality and international interest. For better transparency and understanding of the information to be published, I suggest the following additions:

  1. As long as the study covers cities in China, this should be evident from the title of the article. I propose to add to the title of the article "… in China".
  2. It would be interesting to mention some legislative frameworks in force regarding the optics of the political-administrative environment in the direction of encouraging in obtaining low carbon city and supporting green technology innovation (or an evolution of the legislative perspective, in the interval of years studied).
  3. In some places, there are too many less relevant quotes. For example in lines 77-79: “… based assessment of these policies and found that the promotion of these cities is conducive to reducing carbon emissions [21,23], decreasing energy consumption [22], and optimizing industrial structure [25,26 ]. ” What do you mean when you say that the promotion of these cities is conducive to reducing carbon emissions ?, is not clear ... and in both cited bibliographic sources (21 and 23) reference is made to this aspect, surely?!
  4. Are all the variables in table 1 described in the text (what impact do they, each, in turn, bring to the study)?
  5. The acronym R&D is used 17 times but it might be appropriate to indicate what it represents (Research & Development I assume).
  6. The last column of Tables 2 and 4 is cut and the entered values ​​cannot be read in full.
  7. In subchapter 3.3 (Model Settings) could be indicated the bibliographic source based on which the calculation algorithm expressed in relation (1) was developed, for the validation of the correctness of the model construction.
  8. In subchapter 4.3 you applied the DID model, but in 4.3.1 and in the legend of tables 3 and 4 the acronym PSM also appears, I consider that it would be useful to explain what it means. 9. It should also be explained under tab 4, 5, 6, 7, and 8 which represent the signs *** (footnotes). 10. To be checked in accordance with the requirements of the journal if the footnotes are allowed to be applied as text at the bottom of the pages (such as on pages 5, 6, 8, 10, 14, and 15).

Reviewer 2 Report

This is a report on a study that aims to expand the relevant literature on the low-carbon city pilot policy in China by clarifying the role of low-carbon pilot policy direction and influence in finding that the introduction of the pilot policy was highly conducive to the green technology innovation of enterprises in eastern cities in China and high carbon emission industries.

Four hypotheses were presented:

Hypothesis 1. Low-carbon city pilot policies are conducive to corporate green technological innovation.

Hypothesis 2. Low-carbon city pilot policies are more conducive to corporate green technological innovation in eastern cities than in central and western cities.

Hypothesis 3. Low-carbon city policies are more conducive to green technology innovation in high-carbon emission industries than in low-carbon emission industries.

Hypothesis 4. To implement a low-carbon city pilot policy, enterprises should promote green technological innovation by alleviating financing constraints.

The first three hypotheses were verified by the research analysis while the last hypothesis was found to require further testing of the relevant data to be verified or rejected.

The paper is well-written, clear and examines the data from various perspectives in order to come to conclusions regarding the four hypotheses.

Although the data are well-analyzed, what is not explained is why, if the first batch of low-carbon pilot projects was launched in 2010 (lines 31-32) that the data included green patent data of listed companies from 2005 to 2019 (lines 211-212).  Why include the data from before 2010?  The reason for this should be evident.

Specific comments referring to line numbers and tables.

2-3 “Does the low-carbon city pilot policy promote green technology innovation? Based on green patent data of listed companies” does not read appropriately.  The title should be changed to either “Determining if low-carbon city pilot policy promotes green technology innovation based on green patent data of Chinese A-share listed companies” or “Does the low-carbon city pilot policy promote green technology innovation? A study based on green patent data of Chinese A-share listed companies”.  Chinese A-share  should be added to the title for clarity.

55 Change “increase” to “increasing”.

108 As there four hypotheses, the title should be “Research hypotheses”

133-134 There is no reference to Porter and Vender Linde.  Please provide one.

146 The first time you are mentioning eastern cities, include that they are eastern cities in China.

148 Similarly, the first time you use central and western cities say that they are in China.

293 Table 1. needs to be reworked so that column 1, 4 and 6 are all wide enough for the information to appear on one line.  To make sufficient room to do this, it is suggested that column 2 be made thinner since it takes up two lines currently and more could be put on the second line by making it thinner to accommodate increasing the size of the aforementioned three columns.

352 The size of Table 2. needs to be decreased so that the right hand column is fully visible.  As well, the width of the Indebt row should be increased so that the reported data are not cut off at the bottom.

396 The size of Table 4. needs to be decreased so that the right hand column is fully visible.

525 The size of Table 8. needs to be decreased so that the right hand column is fully visible.

563-567 In line 332, it is mentioned Hypothesis 1 is verified.  In lines 444 and 483, the same is said for Hypothesis 2 and 3 respectively.  However, it is not mentioned that Hypothesis 4 has not been verified.  There needs to be mention of Hypothesis 4 specifically.

605-608 Given that with this study there is no verification of Hypothesis 4, it is premature for the authors to conclude that tax incentives and government subsidies can be used to ease financing constraints.  There are no data that indicated this conclusively.

The citing of both footnotes 1 and 2 is irregular. The other footnotes are cited as they should be, but 1 and 2 need to be standardized. 

Reviewer 3 Report

Dear Authors, 

Thank you for your submission! The study made interesting findings that are worth considering! Green technological innovations are characterized by high uncertainty, high capital requirements, and strong externalities. Firms often face high funding risks. Due to high-profit pressures, these firms focus on short-term returns, which hampers innovation and long-term sustainable investment. To implement a low-carbon urban pilot policy, businesses need to support green technological innovation to ease funding constraints. Richer cities are more successful in implementing green urban development policies than poorer regions. The effectiveness of CO2 reduction is significantly higher in regions with poorer and more polluting industries, but due to the long payback period, companies do not undertake R&D projects. In order to tackle climate change and achieve sustainable development, the green and low-carbon economy have received a new emphasis in China. However, for most of China’s cities, the industrial sector still dominates the industrial structure, and the energy structure of this sector is primarily concentrated on high-carbon energy. The low-carbon development process poses a number of challenges that require the market to play a key role in stimulating the green innovation vitality of businesses.

I think the authors did very interesting and comprehensive research. The methodology and number of samples are also convincing to the reader. Based on the conclusions and based on the suggestions made, I would like to suggest to the authors to look at the literature related to the development of circular business models. The biggest problem with pilot city programs is that the program is not successful in the cities where pollution is highest. Stakeholders see the root of the problem in funding and return risk. The literature related to the development of the circular business model addresses precisely this issue. It does not propose technological innovation, it does not want to implement organizational innovation, but leads the value creation process through business innovation with the lowest investment costs and financing risks. I suggest integrating this circular business section into the literature review and discussion section for the authors.
The dissertation is a very good quality work, I found spelling or misspellings in some places, but their number is not relevant. If you read the article again, you will surely find them in the text. The article is basically long, so in case of further enlargement, I suggest shortening the methodological part by 10-20%.

Best wishes, 

Reviewer 4 Report

1. Please consider whether it is necessary to refer to the same source more than once in the same paragraph, e.g.:

  • "Some studies  either  show  that  such  relationship  is inconsistent [14-15] or point out a non-linear relationship [16-20], and due to differences in their samples and indicators, these studies produce varying conclusions. Yu [18] examined  resource-basedcompanies  and found  an  inverted  U-shaped  relationship  between environmental regulations and corporate green technology innovation performance. Using inter-provincial data, Zhang [19] found that the impact of environmental regulations on  green  technological  innovation  has  a U-shaped  characteristic  of  initially  restraining and then promoting. Wang [20] conducted an empirical analysis by using the green patent data of Chinese listed companies and confirmed the aforementioned U-shaped relationship."
  • "In low-carbon cities, low-carbon city pilot policies are used as comprehensive environmental regulation tools, and the effect of their implementation has received much research  attention. Many scholars have  adopted  synthetic control methods  [21,22]  or conducted  a difference-in-differences  (DID)  [23-26] based  assessment  of  these policies  and found that the promotion of these cities is conducive to reducing carbon emissions [21,23], decreasing  energy  consumption [22],  and  optimizing  industrial  structure  [25,26]."

2. Some Tables do not fit on the page, e.g.:

  • Table 2.
  • Table 4.

3. Please adapt the manuscript to the journal's requirements, e.g.: no dots in the titles of Tables and Figures.

4. Errors in References, e.g. year of publication should be bold and unnecessary characters in items 46 and 48 "— —" 

5. Please shorten the section "Methodology and data" a bit.

6. Some conclusions seem too obvious. Please try to reformulate them a bit, e.g.:

  • "Tax incentives play an important role  in  promoting  the  green technology  innovation  of  enterprises  and  mainly  focus  on green invention patents."
  • "Tax incentives and government subsidies can be used to ease the financing constraints of relevant  enterprises,  promote  their  green technology  innovation,  give  full  play  to  their main role in developing low-carbon industries and clean technologies, and gradually optimize the industrial structure of pilot areas."

Perhaps it is worth indicating specific measures of active fiscal policy or referring to the types of enterprises (e.g. a certain size) that most require support.

Reviewer 5 Report

The paper deals with a very interesting topic and provides a good basis for interesting future research. The paper is original, written in good English, with appropriate methodology and literature review. The structure of the paper is logical, the text is easy to read. The findings are presented and discussed. The aim of the paper was to explore the impact of low-carbon city pilot policies on corporate green technology innovation from multiple dimensions. I like the scientific content of the paper, however there some technical things: please correct Table 2 and 4 (maybe it is better to present them horizontally as now it is not possible to see all the results) - there are some missing columns - which I can not read now.

Congratulation on this interesting article!!!
